# Red Blood Cell Membrane Cholesterol May Be a Key Regulator of Sickle Cell Disease Microvascular Complications

**DOI:** 10.3390/membranes12111134

**Published:** 2022-11-11

**Authors:** Eric J. Niesor, Elie Nader, Anne Perez, François Lamour, Renée Benghozi, Alan Remaley, Swee Lay Thein, Philippe Connes

**Affiliations:** 1Hartis Pharma SA Nyon, 1260 Nyon, Switzerland; 2Laboratory LIBM EA7424, Vascular Biology and Red Blood Cell Team, University of Lyon, 69007 Lyon, France; 3National Institutes of Health, Bethesda, MD 20814, USA

**Keywords:** erythrocyte, membrane cholesterol, HDL, apolipoprotein A1, sickle cell disease, type 2 diabetes

## Abstract

Cell membrane lipid composition, especially cholesterol, affects many functions of embedded enzymes, transporters and receptors in red blood cells (RBC). High membrane cholesterol content affects the RBCs’ main vital function, O_2_ and CO_2_ transport and delivery, with consequences on peripheral tissue physiology and pathology. A high degree of deformability of RBCs is required to accommodate the size of micro-vessels with diameters significantly lower than RBCs. The potential therapeutic role of high-density lipoproteins (HDL) in the removal of cholesterol and its activity regarding maintenance of an optimal concentration of RBC membrane cholesterol have not been well investigated. On the contrary, the focus for HDL research has mainly been on the clearance of cholesterol accumulated in atherosclerotic macrophages and plaques. Since all interventions aiming at decreasing cardiovascular diseases by increasing the plasma level of HDL cholesterol have failed so far in large outcome studies, we reviewed the potential role of HDL to remove excess membrane cholesterol from RBC, especially in sickle cell disease (SCD). Indeed, abundant literature supports a consistent decrease in cholesterol transported by all plasma lipoproteins in SCD, in addition to HDL, low- (LDL) and very low-density lipoproteins (VLDL). Unexpectedly, these decreases in plasma were associated with an increase in RBC membrane cholesterol. The concentration and activity of the main enzyme involved in the removal of cholesterol and generation of large HDL particles—lecithin cholesterol ester transferase (LCAT)—are also significantly decreased in SCD. These observations might partially explain the decrease in RBC deformability, diminished gas exchange and tendency of RBCs to aggregate in SCD. We showed that incubation of RBC from SCD patients with human HDL or the HDL-mimetic peptide Fx5A improves the impaired RBC deformability and decreases intracellular reactive oxygen species levels. We propose that the main physiological role of HDL is to regulate the cholesterol/phospholipid ratio (C/PL), which is fundamental to the transport of oxygen and its delivery to peripheral tissues.

## 1. Role of Membrane Cholesterol and Lipids in RBC Physiology

It has been proposed [1] that during biological evolution, membranes with a particular cholesterol content are selected to perform certain functions and typical cell plasma membranes are defined as having a cholesterol/phospholipid (C/PL) molar ratio from 10 to 30%. Along with myelin, RBC membranes display the unique property of being the cell types with the highest C/PL ratio (around 50%) close to that at which cholesterol crystallization can occur (60–70%) [1]. Because of their localization and the blood–brain barrier, only RBC cholesterol can freely exchange with plasma lipoproteins.

In addition to modulating the activity of specific membrane proteins, the high C/PL ratio of RBC may facilitate gas exchange, as the low oxygen permeability of phospholipid lipid bilayers increases with enrichment of membranes, with cholesterol reaching a plateau between 40 and 50% [2]. Itel et al. [3] reported the first experiments showing that the permeability of cell membranes for CO_2_ appears to be regulated by membrane cholesterol content and membrane protein gas channels over a wide range spanning 2 to 3 orders of magnitude. Subczynski [4] advanced the hypothesis that, in membranes, a high cholesterol content is responsible for creating hydrophobic channels for oxygen transport parallel to the membrane surface. At the same time, a high cholesterol content is also responsible for creating a barrier to oxygen transport across the membrane, which is located near the membrane surface. Thus, transport of small hydrophobic molecules, including oxygen and NO, is facilitated in hydrophobic membrane channels as compared with transport in water.

Increasing plasma membrane cholesterol contents in RBC also resulted in an increased level of extracellular nitrosation following NO exposure. This led to the suggestion [5] that the impact of cholesterol on membrane fluidity and micro-domain structure contributes to the spatial heterogeneity of NO diffusion and signaling.

Buchwald [6] examined the relationship between plasma cholesterol concentration and RBC membrane cholesterol content in hypercholesterolemic subjects. He observed a direct relationship which regulates RBC membrane O_2_ transport and release, and cellular O_2_ availability. He later proposed to increase tissue oxygenation by lowering plasma cholesterol [7]. In 1978, Cooper [8] established the important role of RBC membrane cholesterol and plasma cholesterol in RBC physiology. He wrote: “Thus, red cell membrane C/PL (Cholesterol/Phospholipid ratio) is sensitive to the C/PL ratio of the plasma environment. Increasing membrane C/PL causes a decrease in membrane fluidity, and these changes are associated with a reduction in membrane permeability, a distortion of cell contour and filterability and a shortening of the survival of red cells in vivo”. Sagawa showed that RBC osmotic fragility is highly correlated with the cholesterol content of the cell membrane and C/PL ratio of the cell membrane [9]. Red cells incubated with plasma had decreased membrane cholesterol and increased osmotic fragility. These observations were confirmed by other authors [10,11], who suggested a more subtle effect of cholesterol enrichment on RBC membranes. They observed that cholesterol enrichment decreased the lipid fluidity of the outer membrane leaflet alone, and cholesterol depletion increased the fluidity mainly of the inner leaflet.

RBCs reaching micro-vessels must be flexible to allow enough deformability to traverse vessels with diameters smaller than their own size. A decrease in RBC deformability may compromise oxygen delivery, especially in the microcirculation of peripheral tissue (especially at extremities), increasing the propensity of thrombus formation.

RBC deformability during exercise may be a reversible physiological mechanism [12], while the alterations of RBCs observed in pathological conditions (inflammation, type 2 diabetes (T2D) and SCD) are more likely poorly reversible and may lead to eryptosis or RBC programmed cell death. Among other factors, the lipid-phase separation and cholesterol level definitely influence the molecular arrangement of the plasma membrane [13]. Cholesterol plays a key role in the molecular and mesoscopic organization of lipid membranes and it is expected that changes in its molecular structure (e.g., through environmental factors such as oxidative stress) may adversely affect membrane properties and function [14]. Oxidative stress can lead to the non-enzymatic oxidation of lipids and sterols in the plasma membrane, which is associated with a decrease in the deformability of the cell.

The RBC is particularly susceptible to oxidative stress due to the high content of pol-unsaturated fatty acids in the membrane and the auto-oxidation of the high concentration of iron-containing hemoglobin within the cell. The membrane dipole potential is a sensitive indicator of lipid organization and is dramatically affected by ROS. It has been observed that hydrogen peroxide causes the formation of spectrin–hemoglobin complexes which stiffens the membrane [15]. Several studies have investigated membrane modifications due to oxidation and related changes in membrane rigidity quantified through changes in cell deformability as measured by ektacytometry [16]. Although some lipid peroxidation was observed, these authors consider that the major changes in membrane rigidity were primarily due to formation of spectrin–globin complexes. The oxidation of sterols was investigated by Kulig [17], who identified two families of oxysterols: tail-oxidized sterols, which are mostly produced by enzymatic processes; and ring-oxidized sterols, formed mostly via reactions with free radicals. The former family of sterols was found to behave similarly to cholesterol in terms of molecular orientation, being roughly parallel to the bilayer normal, but still showing increasing membrane stiffness and suppression of its membrane permeability. In contrast, ring-oxidized sterols are quantitatively different from cholesterol, acquiring a tilted orientation, which disrupts the bilayer structure with potential implications for signaling and other processes in the membranes.

## 2. Cholesterol Movements between RBCs and Plasma Lipoproteins

a.RBC cholesterol movement via plasma lipoproteins and ApoA1

The first step of plasma HDL formation is the secretion of its major protein component apolipoprotein A1 (ApoA1) mainly by the liver and intestine. ApoA1, by interacting with the cell membrane transporter ATP-binding cassette A1 (ABCA1), produces the small, poorly lipidated particle often referred to as pre-β1-HDL [18]. Further lipidation by the addition of cholesterol, phospholipids and triglycerides and the esterification of cholesterol with fatty acids, a reaction performed by the plasma enzyme LCAT, will simultaneously enlarge the size of the HDL particles. Notably, initial investigations of cholesterol removal from the cell membrane by HDL and ApoA1 were conducted by Czarnecka and Yokoyama [19] using RBC membranes. They took advantage of the fact that RBC cell membranes lack reactivity to lipid-free apolipoprotein A1 to generate pre-β1-HDL and, thus, allow the study of cellular cholesterol efflux to plasma lipoproteins exclusively by a nonspecific exchange mechanism. Indeed, free apolipoprotein-mediated cellular lipid efflux seems to depend on a trypsin-susceptible cellular surface factor(s) [20] that RBC may lack, being distinct from physicochemical cholesterol exchange reactions between other cell types and lipoprotein (we can assume today that this membrane protein is ABCA1).

In addition, RBCs cannot esterify cholesterol or store cholesteryl esters in lipid droplets as most other cell types (such as macrophages) can; therefore, by using RBCs as donors, we are dealing with a single pool of free cholesterol. RBCs contain the largest body pool of readily exchangeable free cholesterol [21] (6637 μmoL/70 kg), which exchanges twice as rapidly with HDL than with LDL or VLDL, the ApoB-containing lipoproteins. Approximately 50% of circulating cholesterol is carried in RBC membranes and tracer studies in humans [22] confirmed previous data, which indicate that the magnitude of the cholesterol flux through RBCs is comparable to the total efflux of free cholesterol from tissues. These authors did not observe any significant direct exchange between the cholesterol pool of all peripheral tissues (2,4191 μmoL/70 kg) and RBCs [21]. Thus, RBCs play an intermediate role in reverse cholesterol transport: a mechanism allowing the transfer of cholesterol from peripheral tissues to plasma lipoproteins. Cholesterol then bi-directionally exchanges between lipoproteins and RBCs and is eventually delivered to the liver to be metabolized into bile acids and excreted. Hung et al. [23] found that anemia decreased RCT to the feces by over 35% and that transfusion of [3H]-cholesterol-labeled RBCs led to the robust delivery of the labeled cholesterol to the feces in ApoA1-deficient mice, confirming the role of RBCs in reverse cholesterol transport. This concept that RBCs participate in reverse cholesterol transport was first proposed in 1968 by Glomset [24] and supported by more recent publications [25], where it was proposed that the exchange of cholesterol between acceptors and RBCs proceeded via a one-way transport mechanism from ApoA1 to RBCs and via bidirectional exchange between HDL and RBCs, which may act as an intermediate storage unit for cholesterol, thereby participating in cholesterol transport back to the liver.

The concept of “active cholesterol” was first developed by Lange [26], who hypothesized that a fraction of cholesterol can accumulate in membranes, but when the capacity of plasma membrane phospholipids to form complexes is exceeded, it is mostly dissolved in the low-affinity liquid-disordered phase of the bilayer with high activity or a tendency to escape the bilayer and be more easily transferred to HDL.

More recently, Chakrabarti et al. [27] measured RBC membrane cholesterol accessibility using a bacterial cholesterol-binding toxin as a probe and demonstrated that RBC cholesterol in individuals on hemodialysis who experienced an unexplained increase in atherosclerotic risk had significantly higher RBC cholesterol accessibility. RBC cholesterol accessibility appears to be a stable phenotype with significant inter-individual and ethnic variability, which may deserve further investigations given the ethnic differences in the incidence of cardiovascular diseases.

Quantitation of the in vitro exchange of free cholesterol between human RBCs and lipoproteins [28] led to the conclusion that free sterol is the most rapidly exchanging component of the lipoproteins, suggesting that cholesterol is possibly more easily accessible and more loosely bound than the other components of the lipoprotein aggregate. A substantial quantity of unesterified cholesterol of human RBCs exchanges with HDL and LDL during in vitro incubations.

Overall, these results may have consequences for human RBC physiology; Dahlan et al. [29] demonstrated that in humans, infusion of a 10% intralipid mixture at intermediate and high rates induced a significant decrease in RBC C/PL ratio. This change was still present 18 h after the cessation of the high-rate infusion.

b.Lecithin Cholesterol Acyl Transferase (LCAT) and Scavenger Receptor B1 (SRB1) deficiencies

Although poorly lipidated, small HDL particles, such as pre-β1-HDL and HDL3, can remove cholesterol from membranes. The efficiency of this process, especially in RBCs, is markedly increased by further esterification of cholesterol with fatty acids: a reaction performed by the plasma enzyme LCAT [19]. LCAT removes free cholesterol from the HDL surface exchangeable pool and increases the HDL core with highly hydrophobic cholesteryl esters. This process leads to the formation of larger HDL particles (HDL2) that can deliver cholesterol to LDL and VLDL via cholesteryl ester transfer protein (CETP) activity or to tissues expressing the HDL receptor scavenger receptor B1 (SRB1).

Thus, one would expect that deficiency in LCAT or SRB1 may indirectly affect RBC physiology. This is indeed the case in humans with LCAT deficiencies [30] who suffer from major pathologies and defects in RBC maturation and function. LCAT-deficient individuals had abnormally shaped RBCs that were less deformable than normal RBCs under high-shear-stress conditions. The partial depletion of membrane cholesterol from the patient’s RBCs could be achieved by incubation with normal plasma with LCAT activity. Further support for an indirect role of LCAT in RBC physiology and diseases came from Sagawa et al. [9] who showed, in a series of in vitro investigations, that RBCs incubated with plasma had decreased membrane cholesterol and increased osmotic fragility, but the change was prevented by the inactivation of plasma LCAT, suggesting that LCAT activity in plasma is an influential factor in controlling the cholesterol content of RBCs.

LCAT activity is blunted in uremia [31] and occurs with a significant increase in membrane cholesterol and the C/PL molar ratio in RBCs, which may be relevant to the progression of the disease.

In mice, which accumulate HDL cholesterol secondary to deficiency in the HDL receptor SRB1 [32], a 1.7-fold increase in total plasma cholesterol levels was observed, mainly due to a 2.7-fold increase in free cholesterol. The RBC cholesterol content was increased by 1.8-fold, whereas the phospholipid content was not affected, which can result in a decreased deformability of RBCs, thereby leading to an increased risk of damage by shear stress in the microcirculation.

We can conclude that the inefficient removal of RBC cholesterol in LCAT deficiency or the prevention of plasma HDL cholesterol clearance in the absence of SRB1 results in the accumulation of RBC membrane cholesterol to pathological levels with visible morphological and functional changes.

Although the publications cited above strongly support the hypothesis that a primary role of HDL metabolism may be to regulate RBC membrane cholesterol levels, with consequences in common pathological situations such as atherosclerosis and T2D, it has been neglected in favor of investigating the putative link between plasma cholesterol and cholesterol depots in vessel wall atherosclerotic plaques. This link has been demonstrated in animal models of atherosclerosis induced by a high-cholesterol diet and hypercholesterolemia but is unlikely to occur in the natural environment where dietary cholesterol is much lower. In the study performed by Pang et al. [33], the top three food sources of cholesterol were eggs, red meats and seafood. They contributed to more than 90% of the total dietary intake and correlated with plasma cholesterol in the elderly population. There was a substantial urban–rural difference in cholesterol intake. The meta-analysis performed by de Groot et al. [34] including 133,966 subjects from 36 studies confirmed that total and LDL cholesterol levels and triglycerides were consistently higher in residents of urban areas than those of rural areas. Thus, hypercholesterolemia cannot be considered as a natural feature of human physiology.

Indeed, plant-derived diets contain little cholesterol since plants do not synthetize cholesterol but instead contain phytosterols which are chemically and physiologically different from cholesterol. In addition, phytosterols, by competing with cholesterol, prevent cholesterol absorption at the intestinal level in humans, and they are almost totally eliminated afterwards by the liver and bile. In native populations on a low-fat and high-vegetable diet, plasma cholesterol is relatively low and life expectancy is so short that accumulation in cholesterol atherosclerotic plaque is unlikely to occur. In addition, the presence of ApoA1 and high HDL levels in most animal species is not in favor of a major physiological role of HDL in excess arterial cholesterol removal. In the vast majority of mammals, high-density lipoproteins are the predominant class, and may account for up to 80% of the total substances. In humans, the bulk of circulating cholesterol is transported in an esterified form as LDL; however, this is not the case in the majority of mammals, in which HDL is the primary cholesterol carrier [35]. A relatively high LDL and VLDL plasma cholesterol is almost unique to humans, especially those living in an urban environment.

## 3. Low Plasma HDL and High RBC Cholesterol Concentration in SCD and Thalassemia

Sickle cell disease (SCD) is a group of disorders that affect hemoglobin, affecting an estimated 5% of the world population. There are several genotypes, depending on the type of hemoglobin (Hb) synthetized. The most common from is monozygous SCD (i.e., HbSS). It is caused by the presence of an abnormal hemoglobin variant (HbS) in which glutamic acid at the sixth position of the β-globin chain of hemoglobin is replaced by valine. Polymerization of HbS under low-oxygen conditions causes a mechanical distortion of RBCs (sickling). These sickled RBC are poorly deformable and can obstruct normal blood flow in micro-circulation, thereby inducing ischemia in tissues distal to the vascular blockage. Additionally, these poorly deformable RBCs are very fragile with a shortened lifespan, leading to chronic hemolytic anemia. Chronic hemolytic anemia and repeated vaso-occlusive crisis (VOC) underlie the chronic inflammation and vasculopathy in SCD, the basis for pulmonary hypertension, cerebral vasculopathy leading to stroke, osteonecrosis, retinopathy, priapism, leg ulcers, acute chest syndrome and glomerulopathy [36], all contributing to multi-organ damage, impaired quality of life and shortened lifespan. Zorca et al. [36] confirmed significantly decreased plasma levels of total cholesterol, HDL cholesterol, and LDL cholesterol in SCD and demonstrated a significant association between an increased triglyceride/HDL-C ratio and endothelial dysfunction (LDL-C) in SCD versus ethnically matched healthy controls.

It should also be noted that a high concentration of cholesterol in RBCs decreases their capacity to transport oxygen [6,7], which may be detrimental and triggers the sickling process. Several studies reported decreased HDL-C levels in SCD [37,38,39], which may lead to an increased risk for endothelial dysfunction in these patients. This association could be related to the release of oxidized fatty acids during lipolysis, leading to endothelial cell inflammation [40].

Although sharing common mechanisms with atherosclerosis (oxidative stress, inflammation and vascular adhesion), SCD vasculopathy clearly differs in that cholesterol accumulation in the arterial wall and atheroma have not been reported. Surprisingly, while plasma total lipids and cholesterol levels are usually lower in SCD than in healthy individuals [38], levels of total lipids and cholesterol are higher in the RBC membrane of SCD patients compared to controls [41], which could be related to the decreased plasma LCAT levels found in sickle cell anemia (SCA) patients, and more particularly during VOC [42]. Increased RBC membrane cholesterol content in non-SCD patients has also been shown to strongly affect the deformability and visco-elastic properties of the cells [11,43].

Thus, low plasma HDL and low LCAT mass and activity could be the major factors at the origin of increased RBC cholesterol in SCD, and could further negatively impact the already impaired rheological properties of RBCs.

Both SCD and thalassemia are characterized by low plasma HDL levels [44], high RBC membrane cholesterol and low blood levels of lipophilic antioxidants [45]. As early as 1976, RBCs from SCD patients were observed to be significantly enriched in cholesterol compared to those of healthy control subjects. Removing excess cholesterol from RBCs has been achieved using cyclodextrin or HDL, and a pilot study ultimately showed that infusing HDL decreased eNOS and subsequently improved vasodilatation in SCD patients [46].

It has also been reported that levels of ApoA1, the major protein component of HDL, were decreased in SCD [47] and that patients with the lowest ApoA1 levels had blunted flow-mediated dilation in response to acetylcholine and had higher prevalence of pulmonary hypertension [40]. In another study, ApoA1 levels were found to further decrease during VOC in SCA patients [48]. Interestingly, the use of ApoA1 mimetics, such as L-4F, has been demonstrated to improve endothelial dysfunction in SCD mice [49]. Bridging of ApoA1 with the ATP-binding cassette transporter (ABCA1) at the surface of the cells containing an excess of cholesterol also modulates β2-adrenergic responses in endothelial and smooth muscle cells [50]. Thus, it was logically proposed (but never tested) that an excess of cholesterol content in sickle RBC membranes and low ApoA1 levels could result in increased activity of the β2-adrenergic pathway under stressful conditions (increase of epinephrine) and ultimately increase sickle RBC adhesiveness to vascular walls [50].

Increased RBC adhesiveness in SCD also contributes to the initiation of VOC [51,52]. Epinephrine (a major β-adrenergic receptor agonist) has been shown to promote the adhesion of sickle RBCs to laminin through its effects on Lu/BCAM phosphorylation by the adenylate cyclase/cAMP/protein kinase A pathway [53]. Eyler et al. [54] demonstrated that polymorphisms of the β2-adrenergic receptor and adenylate cyclase genes (ADRB2 and ADCY, respectively) affect RBC adhesion to laminin. These authors and others [50] thus proposed that ADRB2 and ADCY polymorphisms could influence SCD severity through the RBC adhesion mechanism. Since membrane cholesterol has been found to enhance the β2-adrenergic signaling pathway, it has been postulated that the high membrane cholesterol level in sickle RBCs might affect the β2-adrenergic response to epinephrine, leading to an increased adhesiveness to endothelial cells.

## 4. Potential Interventions to Regulate RBC Membrane Cholesterol: HDL and ApoA1 Mimetics

Plasma HDL is formed through the interaction between the liver and intestine-secreted ApoA1 and ABCA1 the latter effluxes numerous lipophilic molecules such as cholesterol and phospholipids from cells to ApoA1, thus forming a nascent HDL particle (discoidal pre-β1-HDL particle). LCAT activity further increases the capacity of HDL to take up and transport more lipophilic molecules. This cargo is delivered to HDL receptor (SRB1)-expressing cells and/or exchanged with several tissues including the circulating RBCs, which are in vicinity of the plasma HDL pool.

ApoA1 displays the unique capacity of removing cholesterol from loaded cell membranes through a membrane transporter action (ABCA1 and ABCG1) or by diffusion following the concentration gradient. The phenomenon, also called reverse cholesterol transport, has been very well-documented using cholesterol-labeled macrophages. It should be noted that the large plasma pool of RBC cholesterol has been shown to play an important role in this process.

Therapeutic approaches targeting HDL have been reviewed recently by Uehara et al. [55] and all of them can be considered as potential components of the combination proposed herein. In particular, LCAT activators, CETP inhibitors and modulators, full-length ApoA1, ApoA1-mimetic peptides, such as D4F, L-4F, FAMP and analogs, reconstituted HDL, including ApoA1-phospholipid complexes, and ApoA1 Milano could be considered.

Human recombinant LCAT protein can be infused into animals and humans to increase plasma HDL lipids and apolipoproteins [56]. Similarly, small molecules enhancing LCAT activity and modified LCAT protein with enhanced enzymatic activity are under development; indeed, they increase plasma HDL [57].

The mature sequence of human ApoA1 has a length of 243 amino acid residues and is encoded by exons 3 and 4 of the apolipoprotein gene in chromosome 11. The common lipid-associating motif in ApoA1 is the amphipathic α helix. ApoA1 contains multiple repeats of 22 amino acids (22-mer), each of which is a tandem array of two 11-mers. A large number of peptides mimicking these repeats have been synthetized and display some of the properties of ApoA1, especially with regard to their ability to remove cell membrane cholesterol. For instance, L-4F, an ApoA1 mimetic, dramatically improves vasodilation in hypercholesterolemia and SCD [49].

## 5. Evidence for a Role of HDL and HDL Mimetics on RBC Deformability and Physiology

ApoA1, the major protein of HDL, contains a tandem array of amphipathic helices with varying lipid affinity. Fx-5A is produced by combining sphingomyelin with the 5A peptide, which is a bihelical ApoA1-mimetic peptide in which five hydrophobic amino acids are substituted for alanine in the second helix [58]. Fx5A can efflux cholesterol from labeled macrophages via the ABCA1 transporter. However, a similar effect has not been shown in RBCs. In an initial study, we labeled human RBCs with [^14^C]-cholesterol and then incubated them for 24 h with increasing concentration (0.1 to 100 μg protein/mL) of human HDL or Fx5A. A dose-dependent efflux of labeled [^14^C]-cholesterol was measured after incubation with both HDL and Fx5A (data not shown) confirming that both peptides do remove cholesterol from RBC.

### 5.1. Methods

RBCs were incubated with HDL to measure its effect on deformability. Blood obtained from 7 healthy controls (HbAA) and 12 HbSS patients was collected in EDTA tubes and immediately centrifuged (10 min, 1500× *g* at 20 °C). RBCs were washed in phosphate-buffered saline (PBS) and resuspended in PBS at a final hematocrit of 20%. Then, they were incubated at 37 °C for 40 min with 100 μg/mL of human HDL isolated from human plasma by ultracentrifugation [18]. RBCs were then washed and RBC deformability was measured by ektacytometry [59,60]. RBC deformability, expressed as Elongation Index (EI), was assessed at a shear stress of 3 Pa, which is mainly influenced by elasticity of the RBC membranes, and at a shear stress of 30 Pa, which mainly depends on the intracellular viscosity.

Blood from nine HbSS patients was processed as above and incubated at 37 °C for 40 min with 100 μg/mL and 200 μg/mL Fx5A to assess the effect of the ApoA1 mimetic Fx5A on RBC deformability. RBCs were washed again and deformability was analyzed, as described above, by ektacytometry.

Because SCD RBCs have a high level of ROS [61] and HDL are known to have antioxidant properties, we also investigated the impact of HDL and Fx5A on RBC ROS. For the ROS assay, 7 HbAA and 8 HbSS individuals were included. Blood, collected in citrate tubes, was centrifuged and washed as described above and resuspended at a final hematocrit of 0.4% in PBS. Then, they were incubated at 37 °C for 40 min with 100 μg/mL of human HDL isolated from human plasma by ultracentrifugation or with vehicle (control condition). The washed RBCs were incubated for 20 min with 10 µM of 2′,7′–dichlorofluorescin diacetate (DCFDA), a fluorescent probe sensitive to intra-RBC ROS. The samples were then analyzed by flow cytometry (BD Accuri C6) according to manufacturer’s instructions. The Median Fluorescence Intensity (MFI) of 50,000 gated events was recorded to quantify ROS levels.

To study the effect of Fx5A on RBC ROS, blood from seven HbSS was collected (N = 7) in citrate tubes. RBCs were processed as above and incubated at 37 °C for 40 min with 100 μg/mL of Fx5A or with PBS (control condition). RBCs were washed and incubated for 20 min with 10 uM of DCFDA. Then, the samples were analyzed by flow cytometry, as described above.

Statistical analysis: data are presented as means +/− SEM. The effects of incubation on RBC deformability and ROS levels were analyzed using Student’s *t*-test for paired samples or one-way ANOVA for repeated measures completed by Tukey’s post-hoc tests, when appropriate. Comparisons between HbAA and HbSS subjects were performed using Student’s *t*-test. A *p*-value < 0.05 was set for statistical significance.

### 5.2. Results

#### 5.2.1. Effects of HDL on RBC Deformability and ROS Levels

As expected, RBCs from HbSS patients were less deformable than those of HbAA patients at both shear stress levels (Figure 1A–D). RBC Deformability from healthy volunteers (HbAA) (Figure 1A,B) was not affected by HDL (100 μg/100 mL), whereas RBCs from HbSS patients similarly treated with HDL displayed a marked increase in deformability at both 3 Pa and 30 Pa (Figure 1C,D).

The basal ROS levels were higher by 31.5% ± 10.8% (*p* < 0.05) in RBCs from HbSS vs. HbAA subjects. RBC incubations with HDL did not affect ROS levels in HbAA (Figure 2A) subjects; however, ROS levels were significantly decreased in RBCs from HbSS subjects (Figure 2B).

#### 5.2.2. Effects of Fx5A on RBC Deformability and ROS Levels

The ApoA1 mimetic Fx5A significantly increased HbSS RBC deformability at 3 (*p* < 0.05) and 30 Pa (*p* < 0.01) at both tested doses (100 μg/mL and 200 μg/mL) (Figure 3A,B). A clear dose effect was not observed but the study was not powered for detecting any statistical difference between the lower and higher doses of Fx5A. In addition, the levels of ROS in SCD RBC were significantly reduced by incubating the cells with Fx5A (Figure 4).

In conclusion, both HDL and the ApoA1 mimetic peptide Fx5A can modulate the deformability and level of ROS in RBCs from SCD (SS) patients in vitro. It is noticeable that L4F, another ApoA1 mimetic peptide, has been shown to be beneficial in a mouse model of SCD [49].

Additional studies aiming to quantify the concentration dependence of cholesterol in the membranes of RBCs of SCD (HbSS) patients following incubations with HDL and to determine the concentrations of HDL and Fx5A necessary to normalize RBC membrane cholesterol and RBC deformability are recommended.

## 6. Are Learnings from Sickle Cell Disease Applicable to Type 2 Diabetes and Cardiovascular Diseases?

SCD appears to be characterized by low plasma cholesterol both in HDL and LDL/VLDL lipoprotein fractions with a simultaneous increase in RBC membrane cholesterol, suggesting an inefficient clearance of excess membrane cholesterol. The changes in the RBC membrane very likely affect RBC deformability and ROS levels. Our data suggest that HDL or HDL mimetics may be beneficial to patients suffering from SCD and its microvascular complications.

The role of HDL and therapeutic interventions aiming at raising plasma HDL mainly addressed cardiovascular diseases associated with high plasma cholesterol and atherosclerotic plaque regression. Similarly, the majority of atherothrombosis studies are focused on leukocytes. The demonstration that RBCs can enter the arterial wall during the earliest stages of atheroma and are phagocytosed by smooth muscle cells—a pathogenic mechanism enhanced by hypercholesterolemia, endothelial injury and RBC senescence [62,63]—turned our attention towards erythrocytes.

This review highlights the relationship between HDL formation through LCAT activity and clearance via the SRB1 receptor and RBC physiology and pathology. The potential link between HDL and RBC membrane cholesterol and RBC functions may be exacerbated in SCD and may have implications for more frequent diseases such as T2D and ACS.

Low HDL levels, high RBC membrane cholesterol, high blood viscosity [64] and low blood levels of lipophilic antioxidants may be linked to diabetes [63] and cardiovascular diseases [65,66] and may be at the origin of diabetes complications affecting numerous tissues and organs such as the heart, retina, brain, kidneys and lower limbs [67]. As early as 1982, Cignarelli et al. [68] suggested the possible involvement of increased RBC cholesterol in impairing the microrheological competence of RBCs in diabetes mellitus. Significantly higher RBC cholesterol amounts were found in diabetics compared to controls of similar age; blood filterability values were found to be significantly higher in diabetics and a significant inverse correlation between RBC cholesterol content and serum ApoA1 was also observed. Tsukada et al. [69] and Lee et al. [70] measured RBC deformability and concluded that the deformability of diabetic RBCs was lower than that of RBCs in healthy controls; individuals with abdominal obesity or metabolic abnormalities also exhibit less deformability [71]. Agrawal [72] investigated the role of RBC deformability in T2D with and without diabetic retinopathy and found that the deformability index of RBCs from diabetics, especially those with retinopathy, was significantly lower in comparison with normal healthy controls. In addition to the increase in the RBC C/PL ratio, in the absence of differences in plasma cholesterol, RBC Na/K ATPase activity was significantly increased [73], whereas Mg ATPase was decreased in diabetics.

Li et al. [74] reviewed recent laboratory and computational studies of RBC disorders, such as SCD, hereditary spherocytosis, and diabetes. Alterations to the size and shape of RBCs, due to either mutations of RBC proteins or changes to the extracellular environment, lead to compromised cell deformability, impaired cell stability and increased propensity to aggregate.

Tziakas’s team is one of the most active research groups investigating cardiovascular diseases and RBC membrane cholesterol. They showed for the first time that RBC membrane cholesterol is significantly higher in patients with acute coronary syndrome (ACS) compared with chronic stable angina patients [75,76]. Their findings suggest a potential use of RBC membrane cholesterol as a marker of atheromatous plaque growth and vulnerability and an independent predictor of ACS [77]. In patients with stable coronary disease (CAD), patients’ RBC membrane cholesterol was higher compared to patients with stenosis [66]. Levels of coronary artery scores were correlated positively with membrane cholesterol levels (r = 0.296, *p* < 0.001). In a small group of diabetics with lipid levels normalized with hypolipidemic treatments [78] and in a recent detailed NMR-based lipidomic analysis of RBCs [79], an increase in RBC membrane cholesterol in diabetics was confirmed and insights were provided into molecular lipid features of the membrane microenvironment that influence their vital function and rheological behavior in the microvascular network.

As one would expect, patients with both SCD (or the sickle cell trait) and diabetes have a higher incidence of diabetes-related complications, including retinopathy, nephropathy and hypertension [80,81]. Oxidation of lipids in the vessel wall is an important trigger of atheroma formation. Since RBCs carry both cholesterol and phospholipids in their membranes and the oxidants iron and hemoglobin, their simultaneous presence in infiltrated endothelium could initiate plaque formation. Our review and the abundant literature support the concept that RBC membrane cholesterol is an important physiological and pathological component in the maintenance of vital animal organs through the efficient delivery of oxygen and clearance of CO_2_. Thus, we believe, as proposed by Pavlaki et al. [82], that further research on the interaction of HDL with RBCs is urgently needed to unravel the true role of RBC membrane lipids in vasculopathy, underlying cardiovascular diseases and the complication of SCD and thalassemia with potential impacts on more frequent microcirculatory complications associated with T2D.

## Figures and Tables

**Figure 1 membranes-12-01134-f001:**
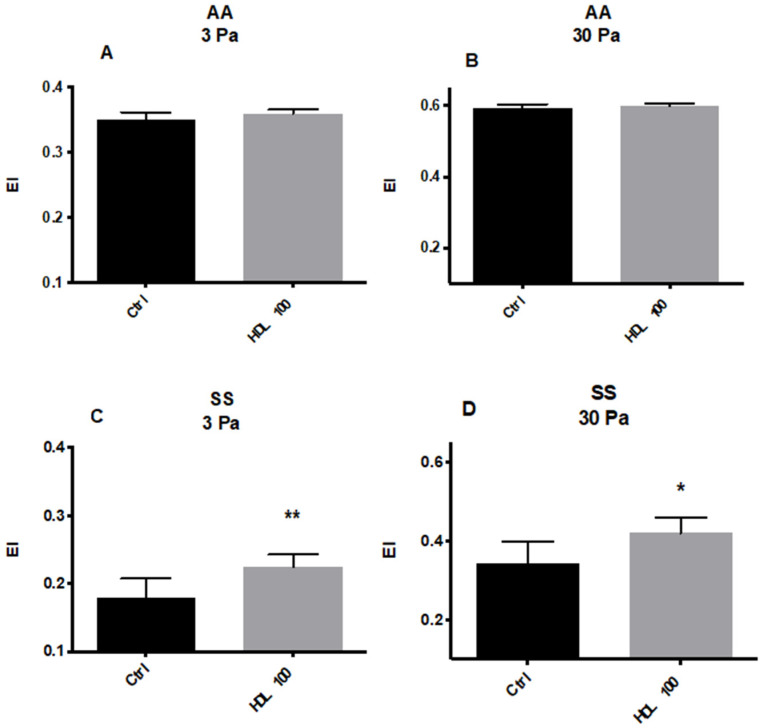
RBC Deformability from Healthy volunteers (HbAA) measured at 3 Pa (**A**) and 30 Pa (**B**) after incubation with human HDL (100 μg/mL). Deformability of RBCs from SCD patients (HbSS) at 3 Pa (**C**) and 30 Pa (**D**) after incubation with human HDL (100μg/mL). *, *p* < 0.05; **, *p* < 0.01. EI: Elongation Index, NS: not significant.

**Figure 2 membranes-12-01134-f002:**
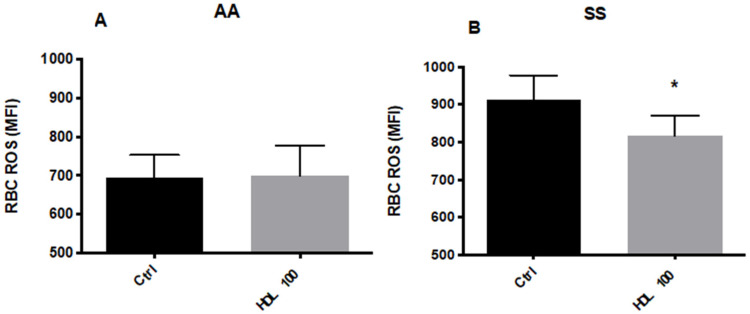
ROS levels in RBCs from HbAA (**A**) and HbSS (**B**) patients after incubation with human HDL (100 μg/mL). NS: not significant; *, *p* < 0.05. MFI: Median Fluorescence Intensity.

**Figure 3 membranes-12-01134-f003:**
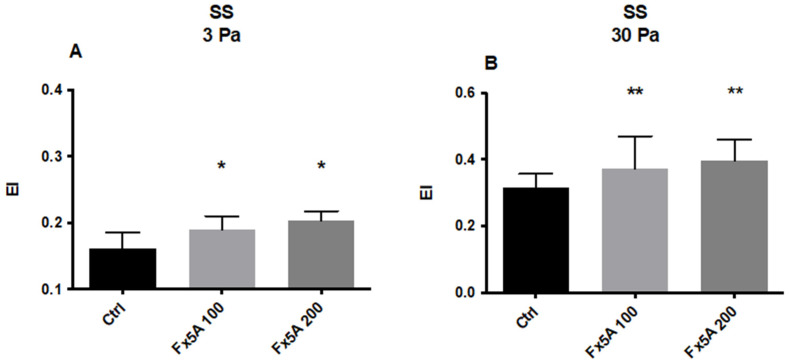
Deformability of RBC from HbSS at 3 (**A**) and 30 Pa (**B**) after incubation with Fx5A (100 µg/mL and 200 µg/mL) (n = 9). *, *p* < 0.05; **, *p* < 0.01. EI: Elongation Index.

**Figure 4 membranes-12-01134-f004:**
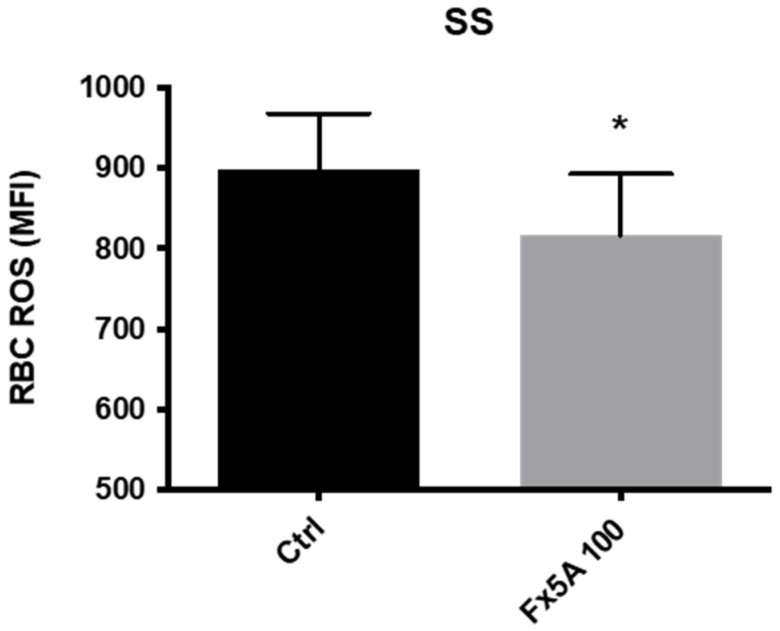
ROS content of RBC from HbSS subjects after incubation with 100 µg/mL of Fx5A (n = 7). *, *p* < 0.05. MFI: Median Fluorescence Intensity.

## Data Availability

Data are available at Lyon University and Hartis Pharma SA.

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
