# Peer review of "Red Blood Cell Membrane Cholesterol May Be a Key Regulator of Sickle Cell Disease Microvascular Complications"

_membranes, 2022, doi:10.3390/membranes12111134_

Round 1
Reviewer 1 Report
This is a very thorough review discussing the role of RBC membrane cholesterol in sickle cell microvascular complications. I have no major comments however there are some minor corrections that would improve the paper.
Minor comments:
1. Having defined Red Blood Cell (RBC) at the beginning, it might be an idea to keep to RBC and not use erythrocyte later in the paper. If you are going to redefine Red Blood Cell in the introduction then this should be done at the first mention “Role of membrane cholesterol and lipids in RBC physiology”.
2. ‘They’ is not defined in this paragraph: “The red blood cell is particularly susceptible to oxidative stress due to the high con-tent of polyunsaturated fatty acids in the membrane and the auto-oxidation of the high concentration of iron containing haemoglobin within the cell. The membrane dipole potential is a sensitive indicator of lipid organisation and is dramatically affected by reactive oxygen species. They observed….”
3. There are a number of typos/language errors that should be corrected. Some listed below:
Title “Red Blood Cell membrane cholesterol may be a key regulator of sickle cell disease microvascular complications”
Abstract “High membrane cholesterol affects their main vital function,….”
“They took advantage of the fact that RBC cell membranes lack reactivity to lipid free apolipoproteins to generate pre-β1-HDL and thus allow study of cellular cholesterol efflux to plasma….”
“RBCs contain the largest body pool of readily exchangeable free cholesterol….”
“This is indeed the case, in humans with LCAT deficiencies 30 who suffer from major pathologies”
“The loss of RBC deformability has been con-sidered to be the primary factor responsible for the vaso-occlusive events, severe hemo-lytic anemia and progressive organ damage in patients suffering….”
“Surprisingly, while plasma total lipids and cholesterol levels are usually lower in SCD”
“the β2-adrenergic receptor and adenylate cyclase 6 genes (ADRB2 and ADCY9, respectively) are affecting RBC adhesion to laminin.”
“all of them can be considered as potential components of the combination proposed herein.”
“Because SCD erythrocytes have a high level of Reactive Oxygen Species (ROS) 61 and HDL are known to have antioxidant properties….”
4. When referring to ‘humans’ or ‘animals’ use the plural.
5. This sentence is not clear: “Overall, these results may have consequences for human RBC physiology as Dahlan et al 29 demonstrated in humans, that RBC lipids can be modified by a single infusion of a lipid emulsion related to the amount of PL liposomes.”
6. Would this all be better as one paragraph? “One would thus expect that LCAT or deficiency in SRB1 may indirectly affect RBC physiology. This is indeed the case, in human with LCAT deficiencies 30 which suffer from major pathologies and defects in RBC maturation and function. Abnormally shaped erythrocytes were less deformable than those of the normal individual under high shear stress conditions. The partial depletion of membrane cholesterol from the patient’s erythrocytes could be achieved by incubation with normal plasma with LCAT activity. Further support for an indirect role of LCAT in RBC physiology and pathology came from Sagawa et al 9 showed in a series of in vitro investigations, that red cells incubated with plasma had de-creased membrane cholesterol and increased osmotic fragility, but the change was pre-vented by the inactivation of plasma LCAT. For this author LCAT activity in the plasma is an influential factor in controlling the cholesterol content of red cells.
LCAT activity is blunted in uremia 31, and occurs with a significant increases in membrane cholesterol and C/PL molar ratio in erythrocytes which may be relevant to the progression of the disease.
In mice, which accumulate HDL-cholesterol secondary to deficiency in the HDL receptor SRB1 32 a 1.7-fold increase in total plasma cholesterol levels was observed, mainly due to a 2.7-fold increase in free cholesterol. The erythrocyte cholesterol content was in-creased by 1.8-fold, whereas the phospholipid content was not affected which can result in a decreased deformability of the erythrocytes, thereby leading to an increased risk of damage by shear stress in the microcirculation.
We can conclude that the inefficient removal of RBC cholesterol in LCAT deficiency or prevention of plasma HDL cholesterol clearance in the absence of SRB1 results in the accumulation of RBC membrane cholesterol to pathological levels with visible morphological and functional changes.”
7. Define ‘SCA’: “which could be related to the decreased plasma LCAT levels found in SCA patients, and more particularly during VOC 42.”
8. g force is normally represented by a lower case g.
9. Specify the ‘vehicle’ used in the control condition.
10. References 12, 14, 33, 60, and 74 are incomplete. Is reference 46 an abstract?
Author Response
Reviewer #1
Comments and Suggestions for Authors
This is a very thorough review discussing the role of RBC membrane cholesterol in sickle cell microvascular complications. I have no major comments however there are some minor corrections that would improve the paper.
Minor comments:
- Having defined Red Blood Cell (RBC) at the beginning, it might be an idea to keep to RBC and not use erythrocyte later in the paper. If you are going to redefine Red Blood Cell in the introduction then this should be done at the first mention “Role of membrane cholesterol and lipids in RBC physiology”.
- Author’s Reply: this has been done throughout the manuscript
- ‘They’ is not defined in this paragraph: “The red blood cell is particularly susceptible to oxidative stress due to the high con-tent of polyunsaturated fatty acids in the membrane and the auto-oxidation of the high concentration of iron containing haemoglobin within the cell. The membrane dipole potential is a sensitive indicator of lipid organisation and is dramatically affected by reactive oxygen species. They observed….”
- Author’s Reply: they was replaced by “It has been observed that hydrogen peroxide causes the formation of spectrin–haemoglobin complexes which stiffens the membrane 15.
- There are a number of typos/language errors that should be corrected. Some listed below:
Title “Red Blood Cell membrane cholesterol may be a key regulator of sickle cell disease microvascular complications”
Abstract “High membrane cholesterol affects their main vital function,….”
“They took advantage of the fact that RBC cell membranes lack reactivity to lipid free apolipoproteins to generate pre-β1-HDL and thus allow study of cellular cholesterol efflux to plasma….”
“RBCs contain the largest body pool of readily exchangeable free cholesterol….”
“This is indeed the case, in humans with LCAT deficiencies 30 who suffer from major pathologies”
“The loss of RBC deformability has been con-sidered to be the primary factor responsible for the vaso-occlusive events, severe hemo-lytic anemia and progressive organ damage in patients suffering….”
“Surprisingly, while plasma total lipids and cholesterol levels are usually lower in SCD”
“the β2-adrenergic receptor and adenylate cyclase 6 genes (ADRB2 and ADCY9, respectively) are affecting RBC adhesion to laminin.”
“all of them can be considered as potential components of the combination proposed herein.”
“Because SCD erythrocytes have a high level of Reactive Oxygen Species (ROS) 61 and HDL are known to have antioxidant properties….”
Author’s Reply: these changes were made
- When referring to ‘humans’ or ‘animals’ use the plural.
- Author’s Reply: done
- This sentence is not clear: “Overall, these results may have consequences for human RBC physiology as Dahlan et al 29 demonstrated in humans, that RBC lipids can be modified by a single infusion of a lipid emulsion related to the amount of PL liposomes.”
- Author’s Reply: this sentence has been modified
- Would this all be better as one paragraph? “One would thus expect that LCAT or deficiency in SRB1 may indirectly affect RBC physiology. This is indeed the case, in human with LCAT deficiencies 30 which suffer from major pathologies and defects in RBC maturation and function. Abnormally shaped erythrocytes were less deformable than those of the normal individual under high shear stress conditions. The partial depletion of membrane cholesterol from the patient’s erythrocytes could be achieved by incubation with normal plasma with LCAT activity. Further support for an indirect role of LCAT in RBC physiology and pathology came from Sagawa et al 9 showed in a series of in vitro investigations, that red cells incubated with plasma had de-creased membrane cholesterol and increased osmotic fragility, but the change was pre-vented by the inactivation of plasma LCAT. For this author LCAT activity in the plasma is an influential factor in controlling the cholesterol content of red cells.
LCAT activity is blunted in uremia 31, and occurs with a significant increases in membrane cholesterol and C/PL molar ratio in erythrocytes which may be relevant to the progression of the disease.
In mice, which accumulate HDL-cholesterol secondary to deficiency in the HDL receptor SRB1 32 a 1.7-fold increase in total plasma cholesterol levels was observed, mainly due to a 2.7-fold increase in free cholesterol. The erythrocyte cholesterol content was in-creased by 1.8-fold, whereas the phospholipid content was not affected which can result in a decreased deformability of the erythrocytes, thereby leading to an increased risk of damage by shear stress in the microcirculation.
We can conclude that the inefficient removal of RBC cholesterol in LCAT deficiency or prevention of plasma HDL cholesterol clearance in the absence of SRB1 results in the accumulation of RBC membrane cholesterol to pathological levels with visible morphological and functional changes.”
- Author’s Reply: these paragraphs have been re-written
- Define ‘SCA’: “which could be related to the decreased plasma LCAT levels found in SCA patients, and more particularly during VOC 42.”
- Author’s Reply: SCA was replaced by SCD
- g force is normally represented by a lower case g.
- Author’s Reply: done
- cify the ‘vehicle’ used in the control condition.
- Author’s reply: PBS was added to the method section
- References 12, 14, 33, 60, and 74 are incomplete. Is reference 46 an abstract?
- Author’s Reply: References were completed and reference 46 is indeed a published poster abstract

Reviewer 2 Report
In their manuscript entitled “Red Blood Cells membrane cholesterol may be a key regulator of sickle cell disease microvascular complications” Niesor et al. have gathered a significant amount of information regarding the relationship between HDL and RBC-membrane cholesterol and their impact to RBC deformability and several disease complications, focusing on the sickle cell background. The authors also present preliminary data regarding the effect of HDL and HDL-mimics upon RBC deformability and ROS accumulation in sickle cell subjects. The manuscript is well structured and contains adequate and meaningful information, but this Reviewer has some comments, as follow:
1. Introduction Section paragraph 1.3: The authors state that SCD affects over 50 million people worldwide, an estimated 5% of the world population. Nonetheless, the world population is around 7.8 billion, thus the percentage they state is not correct. Please correct this discrepancy or rephrase.
2. Methods Section: The authors could write more concisely the assays they performed by firstly describing the two distinct treatments they performed (with HDL and Fx5A) and then describing the deformability and ROS assays for all samples. They could state the number of samples for each category in parentheses, than distinctly presenting the two assays for each comparison.
3. Results Section: In this Reviewer’s opinion, it would be helpful for the reader if Figures 1 and 2 were united in one Figure. In this way the reader will be able to also see the difference between controls and sickle cell subjects, especially since this comparison is stated in the beginning of the Results section.
Author Response
Reviewer # 2
Comments and Suggestions for Authors
In their manuscript entitled “Red Blood Cells membrane cholesterol may be a key regulator of sickle cell disease microvascular complications” Niesor et al. have gathered a significant amount of information regarding the relationship between HDL and RBC-membrane cholesterol and their impact to RBC deformability and several disease complications, focusing on the sickle cell background. The authors also present preliminary data regarding the effect of HDL and HDL-mimics upon RBC deformability and ROS accumulation in sickle cell subjects. The manuscript is well structured and contains adequate and meaningful information, but this Reviewer has some comments, as follow:
- Introduction Section paragraph 1.3: The authors state that SCD affects over 50 million people worldwide, an estimated 5% of the world population. Nonetheless, the world population is around 7.8 billion, thus the percentage they state is not correct. Please correct this discrepancy or rephrase.
- Author’s Reply: we deleted the exact number since indeed the number is changing more than the percentage.
- Methods Section: The authors could write more concisely the assays they performed by firstly describing the two distinct treatments they performed (with HDL and Fx5A) and then describing the deformability and ROS assays for all samples. They could state the number of samples for each category in parentheses, than distinctly presenting the two assays for each comparison.
- Author’s Reply: the method section has been modified accordingly
- Results Section: In this Reviewer’s opinion, it would be helpful for the reader if Figures 1 and 2 were united in one Figure. In this way the reader will be able to also see the difference between controls and sickle cell subjects, especially since this comparison is stated in the beginning of the Results section.
- Author’s Reply: Figure 1 and 2 were combined into a single figure 1 (A,B,C,D) and the y scale is now the same for the different Hb genotypes
